# Association between Spatial Access to Food Outlets, Frequency of Grocery Shopping, and Objectively-Assessed and Self-Reported Fruit and Vegetable Consumption

**DOI:** 10.3390/nu10121974

**Published:** 2018-12-13

**Authors:** Jared T. McGuirt, Stephanie B. Jilcott Pitts, Alison Gustafson

**Affiliations:** 1Department of Nutrition, School of Health and Human Sciences, University of North Carolina at Greensboro, Greensboro, NC 27412, USA; 2Department of Public Health, Brody School of Medicine, East Carolina University, Greenville, NC 27834, USA; jilcotts@ecu.edu; 3Department of Dietetics and Human Nutrition, School of Human Environmental Sciences, University of Kentucky, Lexington, KY 40506, USA; alison.gustafson@uky.edu

**Keywords:** food environment, dietary measurement, Geographic Information Systems, skin carotenoids

## Abstract

Because supermarkets are a critical part of the community food environment, the purpose of this paper is to examine the association between accessibility to the supermarket where participants were surveyed, frequency of shopping at the supermarket, and self-reported and objectively-assessed fruit and vegetable consumption. Accessibility was assessed using Geographic Information Systems (GIS) measured distance and multiple versions of the modified Retail Food Environment Index (mRFEI), including a localized road network buffer version. Frequency of shopping was assessed using self-report. The National Cancer Institute Fruit and Vegetable screener was used to calculate daily servings of fruits and vegetables. Skin carotenoids were assessed using the “Veggie Meter™” which utilizes reflection spectroscopy to non-invasively assess skin carotenoids as an objective measure of fruit and vegetable consumption. Bivariate and multivariable statistics were used to examine the associations in RStudio. There was a positive association between skin carotenoids and the Special Supplemental Nutrition Program for Women Infants and Children (WIC) and mRFEI scores, suggesting that WIC participation and a healthier food environment were associated with objectively-assessed fruit and vegetable consumption (skin carotenoids). Future research should examine these associations using longitudinal study designs and larger sample sizes.

## 1. Introduction

Fruit and vegetable consumption is important for a healthy and balanced diet [1]. Frequent consumption of a variety of fruits and vegetables reduces the risk of diet-related diseases such as heart disease, diabetes, and cancer [2,3,4,5,6]. Despite these benefits, Americans consume less than the recommended amounts of fruits and vegetables [7,8,9,10], with lower consumption levels among rural and disadvantaged populations [11,12]. Numerous interventions have focused on increasing fruit and vegetable consumption by influencing the food environment in various settings, including farmers markets [13,14,15], corner stores [16,17,18,19,20], and supermarkets [21,22,23,24]. Supermarkets, in particular, are the source of the majority of excess calories and sugars in the United States (US) diet [25,26] and are often engineered to promote unhealthy purchases [27,28,29]. Therefore, it is imperative to develop and evaluate supermarket-based initiatives to increase healthy food purchase and consumption. Many interventions have been conducted in supermarkets [21,30,31], yet more research is needed in the supermarket setting, particularly in areas of high obesity prevalence [32,33,34].

Accessibility to healthy food, primarily measured using distance to food outlets such as supermarkets and farmers’ markets, impacts population-level dietary behaviors [35,36]. The importance of geographic access is based on the geographical concept of “friction of distance”, where distance itself hinders interaction between places, and the farther two places are apart, the greater the hindrance, or cost [37]. Thus, accessibility to food outlets that offer healthier items may influence shopping behaviors, dietary behaviors, and nutritional status. However, previous research examining the association between access to supermarkets and dietary quality appears mixed. Cross-sectional studies have found positive associations between access to supermarkets and fruit and vegetable consumption [38,39,40,41] while others have found no association between access to supermarkets and fruit and vegetable consumption [42,43,44,45,46]. Studies examining change in diet after the introduction of a new supermarket have also found mixed effects [47]. 

Overall, these mixed findings could be due to error inherent in self-reported measures of dietary consumption, as the use of self-reported measures of diet can lead to systematic bias or errors in measurement and recall bias [48,49,50]. Additionally, previous studies have used limited approaches to spatial measurement, which may introduce measurement error and model misspecification, including non-road network Euclidean distance and inappropriate distance measures to account for complex travel patterns [51,52,53]. Therefore, the purpose of this paper is to examine the association between accessibility to the supermarket where participants were surveyed, frequency of shopping at the supermarket, and self-reported and objectively-assessed fruit and vegetable consumption.

## 2. Materials and Methods 

### 2.1. Study Settings and Participants

This study was conducted in two supermarkets in two small urban US Office of Management and Budget Metropolitan Statistical areas in the US: Wilson, North Carolina (NC) (population 49,620; Census Designated Urbanized Cluster at least 2500 and less than 50,000 people) and Greenville, NC, (population 91,495; Census Designated Urbanized Area of 50,000 or more). Both areas are located within the US “Stroke Belt” (a geographical region with elevated levels of cerebrovascular accident) and have adult obesity rates in excess of 30%, have 20% of adults reporting fair or poor health, and over 20% of the population being food insecure [54]. Both locations have limited public transportation systems, which limits accessibility to food outlets, particularly for low-income individuals. 

Participants were recruited within two supermarkets (both a part of separate regional chains) during June-November 2017. Shoppers were approached during a variety of outlet times and days following the completion of their purchases before exiting the outlet. Participants had to be aged 18 or older, shopped at the outlet a minimum of three times prior to that shopping trip, purchased at least 5 items during that shopping trip, and spoke English. Participants completed customer intercept surveys, had their skin carotenoids assessed using the Veggie Meter™ and were given a $10 cash incentive upon data collection completion. A waiver of written informed consent was obtained, and each customer was provided with information about the study. Then, interested participants provided verbal consent before completing the study components. The East Carolina University Institutional Review Board approved this study (UMCIRB 16-001167). Participants reported demographic information, including date of birth, age, sex, race, height, weight, marital status, education level, annual household income, number of children, Supplemental Nutrition Assistance Program (SNAP) status, Special Supplemental Nutrition Program for Women Infants and Children (WIC) status, residential physical address, and length of time at their current home residential address. 

### 2.2. Food Access Variables

The physical address of each participant’s residence was collected in the survey. Due to incomplete or missing address data (30% of the original sample), we were able to geocode 134/191 (70%) addresses. The physical addresses for the supermarket where they were shopping at the point of data collation were also geocoded. To understand the influence of the larger food environment, we used the Reference USA (InfoGroup, 2018; Papillion, NE, USA) business database (contains phone verified business listings) to find food stores within the two counties where data collection was taking place and all contiguous counties (to account for boundary effects). We extracted physical addresses for the following store types: area chain supermarkets (North American Industry Classification System (NAICS) 445110; chains with 50 or more employees), grocery stores (NAICS 45110); subcategorized as large (10–49 employees) and small (three or fewer)), supercenters (NAICS 452910), wholesale clubs (NAICS 452910), convenience stores (NAICS 445120), dollar stores (NAICS 452990), fruit and vegetable markets (NAICS 445230) and limited service restaurants (NAICS 722513). Additional cleaning for category misclassification and duplicate outlets was conducted. All residential and business addresses were batch geocoded with the Google Maps application programming interface through the BatchGeo website, and geocoded to the highest level of accuracy possible, either to the rooftop (street address precision) or range-interpolated (interpolated between 2 precise points) levels. Additional verification was made using Google Maps street listings and Google satellite imagery (Google LLC, Mountain View, CA, USA).

The Center for Disease Control and Prevention’s (CDC) modified Retail Food Environment Index (mRFEI) includes a ratio of healthy food outlets (supermarkets, larger grocery outlets, supercenters, and fruit and vegetable markets within census tracts or half a mile from the tract boundary) to less healthy food outlets (fast food restaurants, small grocery outlets, and convenience outlets within census tracts or half a mile from the tract boundary) [55]. The mRFEI was calculated using the geocoded food outlet variables matched to the prescribed outlet healthiness categories, by spatially joining geocoded food outlets to study area Census tracts (which were given half-mile buffers around the census tract boundary as prescribed to account for boundary effects). 

The distance from a participant’s home address to the supermarket where the participant was surveyed, was calculated using the Environmental Systems Research Institute (ESRI) (Redlands, CA, USA) ArcGIS online network analysis services, which uses road networks to simulate typical travel behaviors. Spatial variables were calculated using North America-wide street network data to increase accuracy and reduce edge effects by accounting for customers’ ability to traverse administrative (i.e., county) boundaries. A time variable was also generated, which indicated the typical time needed to traverse the network using predictive traffic modeling, which takes typical conditions into account [56]. Service area analyses were generated to produce 1, 3, 5 and 10-mile road network buffers from each participant’s home address. These buffers were then spatially joined to geocoded food outlet data to determine food outlet counts within the buffer area. From this, we generated a localized mRFEI Service Area in 1, 3, and 5 mile buffers from home for each participant in order to have a more localized mRFEI score compared to the typical census-tract level score. Participants were also asked the perceived distance from their home to the outlet in miles and/or blocks. 

### 2.3. Food Outlet Shopping

Participants were asked how often they shop at the outlet where they were surveyed, which included options that ranged from “this is my first visit” to “six to seven days a week”. They were also asked the reasons why they shopped at the outlet, and the options they were given regarding accessibility included: (1) whether it was close to where they live, (2) worked, or (3) went to school, or (4) if it was on their way to work, home or school. Other options they were given included: (1) liking the food offered, (2) the store sells healthy foods, (3) they meet friends at the store, (4) the store has good prices/is inexpensive, and (5) the store has good quality. 

### 2.4. Fruit and Vegetable Consumption

The questionnaire assessed consumption of fruits and vegetables over the past thirty days using the National Cancer Institute’s (NCI) “All Day” Fruit and Vegetable Screener (NCI F&V) [57]. Following the participant’s responses in reference to their fruit and vegetable consumption over the past thirty days, the researcher asked questions regarding the participant’s typical eating habits, based on frequency and quantity. The NCI Screener contains questions pertaining to frequency and amounts for an established set of food items, including: 100% Juice (orange, apple, grape, or grapefruit); Fruit (fresh, canned, frozen, no juice); Lettuce Salad; French Fries or Fried Potatoes; Other White Potatoes (baked, boiled, and mashed potatoes, potato salad, and white potatoes that were not fried); Cooked Dried Beans; and Other Vegetables; and Tomato Sauce (tomato sauce on pasta or macaroni, rice, pizza and other dishes). There were also questions regarding the frequency of Vegetable Mixtures (foods such as sandwiches, casseroles, stews, stir-fry, omelets, and tacos). To calculate the daily servings of fruits and vegetables from the screener, we used the standard suggested algorithm, as provided on the NCI website [57]. 

To objectively measure fruit and vegetable consumption, participants’ skin carotenoids were measured using non-invasive reflection spectroscopy (“Veggie Meter™”), developed by Longevity Link Corp., Salt Lake City, Utah. The Veggie Meter™ assesses the optical density of skin carotenoids [58,59]. The Veggie Meter™ was found to be a valid measure of fruit and vegetable consumption among a racially diverse sample in eastern North Carolina [60], with a correlation coefficient of 0.71 (*p* < 0.0001) between plasma carotenoids assessed via High Performance Liquid Chromatography (the gold standard) and skin carotenoids assessed via the Veggie Meter™. The Veggie Meter™ provides a score for skin carotenoids on a scale from 0–800, with a higher score indicating more skin carotenoids. Each participant had his/her skin carotenoids measured three separate times, and values were averaged.

### 2.5. Statistical Analysis

Our statistical approach followed traditional recommendations of building up from univariate to bivariate, to multivariable analysis using pre-determined variables of interest, which have face validity, are logical and supported by the literature. Summary statistics, including means and standard deviations, were generated for demographic variables, self-reported fruit and vegetable consumption, objectively-measured skin carotenoids, and objectively measured Geographic Information System (GIS) distances and mRFEI. Proportions were generated for categorical variables of interest, including sex, race, participation in SNAP and WIC, and income. We assessed whether the outlet they shopped at was the closest supermarket to their home. 

Bivariate analyses were conducted, including *t*-tests and Mann–Whitney U-tests to measure differences in skin carotenoid levels and distance to the supermarket (in miles and minutes) across categorical variables that could be associated with the primary dependent (skin carotenoid levels) and independent (distance to supermarket shopped) variables of interest, including: race (white non-Hispanic/non-white), receiving SNAP (yes/no), receiving WIC (yes/no), annual income less than $40,000 (yes/no), high school education or less (yes/no), and reasons for shopping at the supermarket. Differences in average skin carotenoids were also examined across frequency of shopping categories (“A few times per year”, "Once month or less", “Once every 3 weeks”, “Once every 2 weeks”, “Once per week or less”, “Once per week”, “Two to three times per week”, “Four to five times per week”, and “Six to seven times per week”).

We examined bivariate associations between objectively-measured fruit and vegetable consumption (Veggie Meter™) and spatial access variables: road network distance and time to supermarket where surveyed; mRFEI, number of mRFEI-defined healthy outlets, number of mRFEI-defined less healthy outlets, density of supermarkets around residence at 1, 3, and 5 mile localized Service Areas (chosen because previous research indicates that buffers 1 mile and larger are likely more meaningful in influencing dietary behaviors and because they are reasonable distances to travel for food given variation in distance from commercial areas) [53], and distance to the closest supermarket and supercenter, using Spearman’s correlation coefficient (rho). Due to curvilinear correlations observed in scatterplots, and based on the literature and knowledge of the topic, the research team thought that various subgroups might be influencing the relationship between primary variables of interest (skin carotenoids and distance to the supermarket shopped). These subgroups were examined, including by site location (Outlet A versus Outlet B), frequency of shopping, SNAP or WIC participation, and reasons for shopping at the outlet. When examining the variable “distance to supermarket where surveyed”, we removed those participants that stated that they shopped at that supermarket because it was close to where they worked, with no indication they shopped there because it was close to home. We excluded these participants because distance variables were all measured from the home address, and therefore, their use could bias expected associations with dietary outcomes. 

Unadjusted linear and polynomial regression (quadratic, cubic, log, inverse(1/x)), and inverse (1/sqrt(x)) analyses were performed for all spatial independent variables to determine the best model fit, given the curvilinear relationships found in the bivariate analysis. Model fit was determined by the multiple r-squared value for each regression (the proportion of the variance for a dependent variable that is explained by the independent variable). We reported each of the models with the best fit. The primary dependent variable was objectively-measured fruit and vegetable consumption (Veggie Meter™) and the primary independent variable of interest was road network distance (in miles) to the supermarket where the customer was shopping during survey collection. Other independent variables of interest included mRFEI, the number of mRFEI healthy outlets, number of mRFEI less healthy outlets, 1, 3, and 5 mile localized Service Area road network buffer mRFEI scores, density of supermarkets around residence at 1, 3, and 5 miles, distance to the closest supermarket, shopping at the outlet because it was near their home, WIC receipt (yes/no), and SNAP receipt (yes/no). Covariates included age (in years), sex (male or female), and race/ethnicity (non-Hispanic white/non-white). We also assessed the data for clustering based on statistically significant variables associated with skin carotenoids using the Hopkins statistic but found a low clustering tendency in the dataset (Hopkins: 0.19). All analyses were conducted in RStudio Version 1.1.419 (RStudio Team (2016). RStudio: Integrated Development for R. RStudio, Inc., Boston, MA, USA; URL http://www.rstudio.com/).

## 3. Results

### 3.1. Participant Characteristics 

A summary of the participants’ sociodemographic and health-related characteristics is found in Table 1. Participants were mostly female (75.7%) and African American (86.0%). The mean age of participants was 46.4 years old, 37.5% had a high school education or less, and a majority (74.2%) of participants had an annual household income of less than $40,000. Thirty-nine percent were SNAP participants, and 9% were WIC participants. Mean BMI (calculated from self-reported height and weight) was 32.8 (standard deviation (sd) = 7.9) kg/m^2^. 

The mean skin carotenoid score was 250.5 (range = 11.33–450.3, sd = 75.4). Mean fruit consumption was 1.5 (sd = 2.1) servings/day, mean total vegetable consumption was 1.8 (sd = 1.9) servings/day, and mean total vegetable consumption minus fried potatoes was 1.7 (sd = 1.7) servings/day. 

Participants most commonly reported shopping at the outlet 2–3 times per week (34.8%), with no first-time shoppers and 9 participants (7.0%) shopped at the outlet 1 time or less per month. For 70.6% of participants, the supermarket they were shopping at was not the supermarket that was closest to their home. However, 86.7% stated that a reason they shopped at the supermarket was because it was close to where they live, which was the most mentioned category over “shopped there because it was on their way to home/work/school” (55.1%), “near where they worked” (35.2), and “near where they go to school” (8.0%). Few participants stated they shopped there because it was close to work and not close to their home (*n* = 6), and because it was exclusively on their way to work (*n* = 5). Overall, accessibility to their home was second in the top five reasons for shopping at the supermarket. The top five most frequent reasons, in order of the number of responses, were the following: (1) ‘It has good prices/is inexpensive’ (*n* = 115), (2) ‘It is close to where I live’ (*n* = 110), (3) ‘It sells the food I like’ (*n* = 109), (4) ‘It has good quality’ (*n* = 106), (5a) ’It has good service/You know the owner/staff are friendly and helpful’ (*n* = 104), and (5b) ‘It is clean’ (*n* = 104).

The mean GIS-assessed distance traveled from participants’ residential addresses to the supermarket at which they were surveyed was 4.0 miles (range = 0.07–50.8; sd = 7.7), and the mean GIS-assessed travel time was 9 min (median 6.4, range = 0.63–59.8, sd = 9.7). The mean distance to their closest supermarket was 1.2 miles (range = 0.15–30.0; sd = 3.3), and the mean travel time was 4.6 min (range = 0.6–49, sd = 5.2). 

The mean mRFEI score was 9.9 (range = 0–100; sd = 12.4), indicating that on average, 9.9% of the food retailers within participants’ census tracts are healthy. Participants had an average of 3 (range: 0–10; sd = 2.9) mRFEI-defined healthy outlets, and an average of 21 (range = 0–48; sd = 12.0) mRFEI-defined unhealthy outlets in their census tract.

Those receiving WIC had a higher average skin carotenoid score when compared to those not receiving WIC (284 for WIC participants versus 247 for non-participants, *p* = 0.04), but there was no statistically significant difference in average skin carotenoids for those receiving vs not receiving SNAP (*p* = 0.14), race (black versus white) (*p* = 0.20), those with annual incomes less than $40,000 versus those with higher incomes (*p* = 0.52), or high school education or less versus more than high school (*p* = 0.57). There was no statistically significant difference in skin carotenoids by frequency of shopping (*p* = 0.52), including shopping once or more per week (*p* = 0.86). There was a statistically significant difference in skin carotenoids by gender (*t* = 3.1197, degrees of freedom (df) = 65.582, 95 percent confidence interval: 14.84441–67.63785; *p*-value = 0.003), with males (mean = 282.1) having higher average scores than females (mean = 240.9).

### 3.2. Association between Distance to Supermarket and Dietary Consumption

A summary of the correlations between dietary consumption and spatial access variables is found in Table 2. There were non-statistically significant weak relationships between skin carotenoids and all distance measures. The scatterplot (with polynomial trend and confidence interval) between skin carotenoids and ‘Distance to supermarket where surveyed(miles)’ revealed low skin carotenoid scores at closest proximity, rising to peak around 2 miles, and then decreasing as distance increased (see Figure 1a). The scatterplot between the skin carotenoid score and ‘Distance to supermarket where surveyed (min)’ also suggested lower skin carotenoids scores at closer proximity, reaching a peak around 7 min, and then generally decreasing again. The strongest correlations with the skin carotenoid score were with mRFEI score (*c* = 0.13; *p* = 0.12) and having a supermarket within 10 road network miles (*c* = 0.12; *p* = 0.16). The strongest correlation overall was between self-reported fruit consumption and ‘distance to supermarket where surveyed’ (*c* = −0.30; *p* < 0.05) (Table 2).

### 3.3. Association between Skin Carotenoids and Accessibility to Healthy Food Outlets. 

A summary of associations with simple regression is found in Table 3 (unadjusted). In unadjusted models, distance to the supermarket where surveyed was inversely associated with skin carotenoid score, but not statistically significant (*p* = 0.13). The only spatial variables that had a statistically significant association with the skin carotenoid score was mRFEI (*p* = 0.02), mRFEI 1 mile (*p* = 0.003), and mRFEI 3 mile (*p* = 0.05), which had a positive association indicating that skin carotenoid scores increased in a food environment with a higher ratio of healthy outlets to unhealthy outlets. 

A multiple variable linear regression model was used to examine adjusted associations between skin carotenoids and independent variables of mRFEI Localized Service Area 1 and 3 miles (census tract level mRFEI was excluded due to collinearity with mRFEI Localized Service Area 3 mile), and including covariates of WIC, SNAP, store ID, reasons for shopping, distance to supermarket (inverse) in miles, age, gender, and race (see Table 4). The food environment variable of mRFEI Local Service Area 3 mile was significantly positively associated with skin carotenoid score ((*p* = 0.003), indicating that a healthier ratio of retail foods outlets was related to increased skin carotenoids. Gender was significantly negatively associated (*p* = 0.01), suggesting that males were more likely to have higher skin carotenoid scores in this sample. WIC was significantly positively associated (*p* = 0.03), suggesting that WIC participants were more likely to have higher skin carotenoid scores. 

## 4. Discussion

We found a positive association between skin carotenoids and mRFEI (including mRFEI Localized Service Area 3 miles) from home in the adjusted analysis, indicating that long-term fruit and vegetable consumption may be influenced not simply by proximity to a preferred supermarket, but also by the ratio of healthy to less healthy food retail outlets in the proximal food environment. This finding is consistent with previous research, which found county-level obesity was associated with RFEI (food swamps) versus proximity to supermarkets (food deserts) [61]. 

Less healthy food environments, which include a higher ratio of fast food and convenience outlets relative to supermarkets and farmers’ markets, may encourage consumption of less nutrient dense food items, and reduce and replace purchase and consumption of fruit and vegetables. Less healthy food environments may also trigger food behaviors based on appetite and not physiological hunger cues [62]. This is consistent with previous research suggesting that away from home meals are associated with reduced fruit and vegetable consumption [63,64]. This concept may also explain why access to supermarkets was not consistently associated with healthier dietary behaviors, as microenvironments within supermarkets could be considered less healthy environments, triggering negative food behaviors for those exposed [65]. Given this, our findings may support the use of policies that encourage or create a healthier mix of food stores in the food environment (considering the 3-mile localized road network buffer used in this study), or modifications within the supermarket environment to increase healthier purchasing behaviors, including fruit and vegetable consumption. Additionally, policy efforts aimed at improving food access should consider our findings regarding supermarket choice, where shoppers did not always shop at the supermarket closest to their home. Thus, more careful and complex consideration should be made when placing new stores in the food environment. In adjusted analyses, we found positive associations between WIC and skin carotenoids, consistent with prior findings that WIC has a positive influence on dietary behaviors [66,67]. Our findings also suggest that the impact of distance on fruit and vegetable consumption may have been reduced with WIC participation. These findings support the potential benefit of WIC participation in maintaining diet quality despite reduced accessibility. The fact that WIC had a positive relationship with skin carotenoid scores and SNAP had a negative (though not statistically significant) relationship may be partially explained by the healthier and more restricted food options allowed for purchase under the WIC program, which may encourage more fruit and vegetable consumption [68]. This aligns with recent research which found low diet quality for SNAP participants compared to income-eligible nonparticipants and higher-income individuals, potentially due to increased levels of processed meat consumption [69]. There was a small number of WIC participants (*n* = 12) in the current study, so future studies should investigate this association in a larger sample.

In our study, the two supermarkets where customers were recruited are discount supermarkets located in commercial areas in proximity to lower-income areas, which may potentially explain why immediate proximity may be related to lower consumption, with a rise in consumption between 3–5 miles and 5 to 7 min, and subsequent decrease, then leveling as distance increased. There has been considerable research indicating those with lower incomes but living in neighborhoods with middle to higher income block groups, tend to have better health outcomes and report better dietary intake [70]. Given that these customers were selected from similar stores in similar neighborhoods, the effect that distance has on dietary intake may signify the neighborhood effects rather than simply access to a supermarket. Thus, consideration of the geographic context of the stores being sampled, including sociodemographic factors, may be important in future research.

Most of the participants in this study were shopping at a supermarket that was not the supermarket that was closest to their home, which is similar to previous research findings [43,44,71,72]. However, most participants stated that a reason they shopped at the supermarket was because it was close to where they live. Overall, proximity to the home was the second most prominent factor behind store price among the reasons for shopping at the store, and shoppers also frequently mentioned the factors of ‘selling food I like’, “it has good quality’, “good service’, and ‘it is clean’. This suggests that proximity of the outlet to the home matters, but other factors might influence outlet preference among those that are proximal. This combination of price and accessibility has been found to be a prominent factor in food shopping, particularly for lower-income individuals, in previous research [73,74,75]. The prominence of price may also partially explain the skin carotenoid levels among the participants in this study if cost-conscious shoppers deem fruits and vegetables too expensive relative to other foods. The importance of non-spatial factors may also explain why those closest to the supermarket where they shopped had the lowest skin carotenoid levels. 

Most of the other prominent factors, expressed nearly as frequently as price and distance, may best be categorized under the domain of “acceptability”, defined as “people’s attitudes about attributes of their local food environment, and whether or not the given supply of products meets their personal standards”, or the domain of “accommodation”, defined as “how well food sources accept and adapt to resident’s needs” [36]. The concept of ‘cultural capital’, which refers to how the subjective aspects of the store, including ‘climate’ or ‘mood’ align with the customer’s self-image, may also influence shopping decisions [76]. Research has found that supermarkets are class-stratified, where shoppers from different social classes shop at the outlet that caters to their needs, suggesting that customers want a differentiated environment where they are around customers who resemble themselves [76,77]. Perceptions of store level ‘acceptability’ may be a prominent factor determining store and item choice. For example, research looking at supermarket shopping and social class has found an “isn’t for me” perception [76]. Measurement of ‘acceptability’ and ‘accommodation’ of the food environment has been limited and underdeveloped in the literature, thus further research is needed [36].

Our findings suggest a high level of complexity in the relationship between the food environment and fruit and vegetable consumption, as displayed in the curvilinear relationship between the two variables, which appears to possibly be explained by the combined influence of objective and subjective/attitudinal factors. Consumers did not typically shop at the supermarket closest to their home, but were influenced by a more complex food environment composition as indicated by mRFEI, and by concepts like ‘acceptability’, ‘accommodation’, and ‘cultural capital’. Access to a supermarket also did not necessarily lead to healthier behaviors, which could be the result of exposure to store level, less healthy micro-environments or residing in neighborhoods with an overall lower socioeconomic gradient. Given these findings, existing measurement approaches, including the mRFEI, may be too simplistic given the complex nature of food shopping and food purchasing decisions. There is a need for a more complex and hyper-contextual composite measure of the food environment that factors in objective measures of the food environment (distance, food store mix, price, availability) along with more subjective measures that can capture concepts like ‘acceptability’, ‘accommodation’, and ‘cultural capital’. To support that, there may be a need to diversify the way we conceive and measure the built food environment, taking into account the physical composition of the retail environment where the food store resides (e.g., retail density), localized sociodemographic variables (e.g., age, income), and traffic patterns. Similarly, there is a need to diversify the way that subjective/attitudinal factors are measured to better capture this important component. With the vast availability of secondary data now available to examine these hyper-contextual factors, efforts should be made to determine how to best obtain and aggregate the data and then configure a composite score that factors in all of these measures. 

In our sample, men had higher skin carotenoids levels than women. This finding is inconsistent with previous research suggesting that women have higher skin carotenoid status than men based on typically higher rates of fruit and vegetable consumption and smaller body size [78], and elevated plasma carotenoid levels due to physiological differences between the genders [79]. One possible reason for our findings was a statistically significant lower BMI (*p* = 0.006) for males (mean = 29) versus females (mean = 34) in our sample. Previous research has found a negative association between BMI and total plasma carotenoid level concentration due to storage/accumulation of carotenoids in adipose tissue [79,80]. Skin and plasma carotenoid levels have been found to correlate, although skin may be a longer-term measure due to increased rates of depletion in plasma [81]. This finding should be further explored in future research. 

We found no statistically significant differences in skin carotenoid levels across levels of frequency of shopping. Previous research in a similar population also found no significant associations between frequency of shopping and fruit and vegetable consumption for both supermarkets and farmers markets [82]. Less frequent shopping would seem to be an indicator of less frequent fruit and vegetable consumption due to the limited shelf life and increased rate of spoilage of produce, but it may be that less frequent shoppers are compensating for increased distance to the store by purchasing frozen and canned fruits and vegetables. A high percentage of the sample shopped at the supermarket more than once per week, so it could be that our sample had insufficient representation from less frequent shoppers. Future research should further explore this finding. 

The strengths of the study include the objective measurement of dietary intake, the use of multiple methods of assessing dietary intake, assessing perceived and objective measures of food access, and using multiple approaches to spatially assess food environment access. Study weaknesses include the cross-sectional design, small study population, convenience sample, and shoppers were only recruited from two supermarkets in one geographic region, so results may not be representative of other populations. 

## 5. Conclusions

To our knowledge, this is the first study to examine the associations between objectively- measured food environment measures and objectively-measured fruit and vegetable consumption using assessment of skin carotenoids with the validated Veggie Meter™. Our findings suggest that WIC participation and mRFEI are associated with an objective measure of fruit and vegetable consumption (skin carotenoids assessed via the Veggie Meter™) among a sample of supermarket customers. Further research using the Veggie Meter™ to understand supermarket customer dietary habits, including an examination of the effectiveness of supermarket interventions to increase FV consumption, may be warranted. Future research should further examine the relationship between spatial food access variables and objective measures of fruit and vegetable consumption in different populations, with consideration of potentially important subgroup and contextual factors. 

## Figures and Tables

**Figure 1 nutrients-10-01974-f001:**
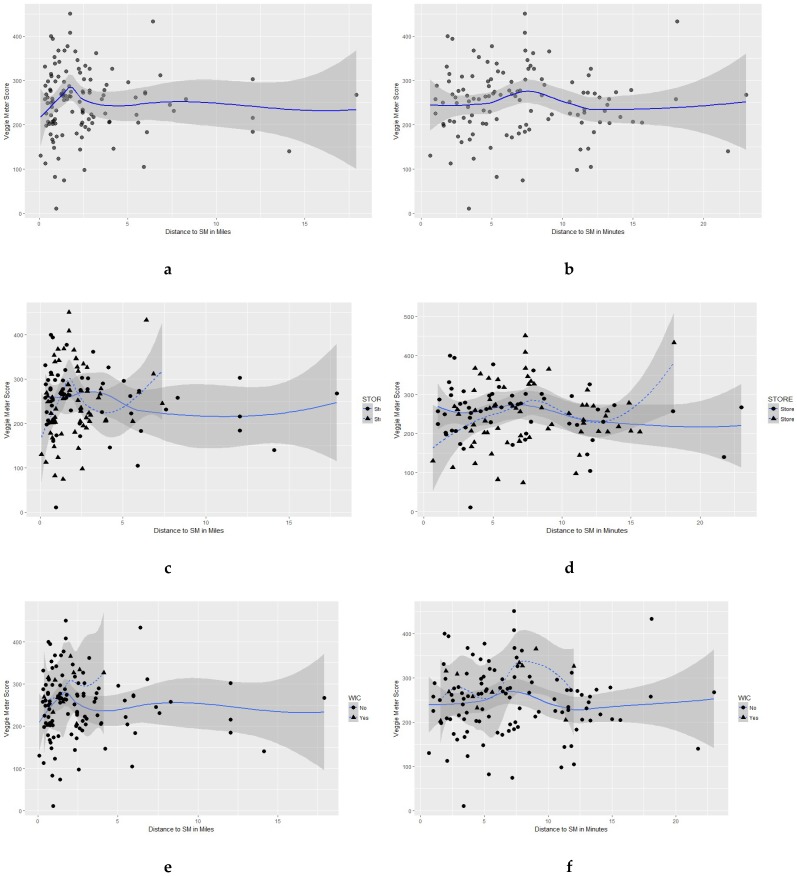
Scatterplots displaying the correlation (with polynomial trend and confidence interval) between skin carotenoid score (mean) and distance to supermarket where surveyed in (**a**) miles and (**b**) min, by store type (**c** (miles)**, d** (min)), WIC (**e** (miles)**, f** (min)) and SNAP (**g** (miles)**, h** (min)).

**Table 1 nutrients-10-01974-t001:** Participant sociodemographic and health-related characteristics (*n* = 136).

Characteristic	Mean (SD) or Number (%)
Sex	
Males	33 (24.2%)
Females	103 (75.7%)
Age (years)	46.4 (15.0)
Race	
Black/African American	117 (86.0%)
White	14 (10.3%)
Other	5 (3.7%)
Receive SNAP	53 (39.0%)
Receive WIC	12 (9.0%)
Education High School or less	51 (37.5%)
Annual Income (<$40,000)	101 (74.2%)
Children in Household	1.2 (1.6)
Frequency of Shopping at outlet (once per week or more)	114 (84.4%)
NCI Fruit, servings per day	1.5 (2.1)
NCI Vegetables, servings per day	1.8 (1.9)
BMI (kg/m^2^)	32.8 (7.9)
Skin Carotenoids, RS Device	250.5 (75.4)
GIS distance to supermarket from home to supermarket surveyed (miles)	4.0 (7.7)
GIS distance to supermarket from home to supermarket surveyed (min)	9.0 (9.7)
GIS distance to closest supermarket (miles)	1.2 (3.3)
GIS distance to closest supermarket (min)	4.6 (5.2)
mRFEI	9.9 (12.4)
mRFEI healthy stores	3 (2.9)
mRFEI less healthy stores	21 (12.0)

SNPC: Supplemental Nutrition Assistance Program; WIC: Special Supplemental Nutrition Program for Women, Infants, and Children; NCI: National Cancer Institute; BMI: Body Mass Index, calculated from self-reported height and weight; RS: Reflection Spectroscopy; GIS: Geographic Information System; mRFEI: modified Retail Food Environment Index.

**Table 2 nutrients-10-01974-t002:** Spearman’s rho and *p*-values for correlations between spatial variables and objectively-assessed and self-reported fruit and vegetable consumption.

Food Environment Measure	RS Device Fruit and Vegetable Consumption	Mean Vegetable Consumption Self-Reported	Mean Vegetable Consumption-(Minus Fried Potatoes) Self-Reported	Mean Fruit Consumption, Self-Reported
**Distance to supermarket where surveyed (miles)**	0.01; *p* = 0.89	−0.17; *p* = 0.12	−0.15; *p* = 0.14	−0.30; *p* = 0.002
**Time to supermarket shopped (min)**	−0.007; *p* = 0.94	−0.16; *p* = 0.13	−0.15; *p* = 0.15	−0.30; *p* = .004
**Distance to closest supermarket**	−0.01; *p* = 0.87	0.07; *p* = 0.51	0.09; *p* = 0.39	−0.13; *p* = 0.19
**Distance to closest supercenter**	0.07; *p* = 0.43	−0.10; *p* = 0.34	−0.09; *p* = 0.43	−0.20; *p* = 0.06
**mRFEI**	0.13; *p* = 0.15	−0.09; *p* = 0.38	−0.12; *p* = 0.25	−0.05; *p* = 0.64
**mRFEI Healthy Outlets**	0.02; *p* = 0.76	−0.02; *p* = 0.89	−0.05; *p* = 0.66	0.0007; *p* = 0.99
**mRFEI Unhealthy Outlets**	−0.07; *p* = 0.43	0.11; *p* = 0.29	0.11; *p* = 0.29	0.08; *p* = 0.45
**mRFEI Service Area 1 mile**	−0.11; *p* = 0.24	−0.05; *p* = 0.67	−0.06; *p* = 0.61	0.07; *p* = 0.48
**mRFEI Service Area 3 mile**	−0.09; *p* = 0.33	0.002; *p* = 0.98	0.01; *p* = 0.89	0.03; *p* = 0.79
**mRFEI Service Area 5 mile**	−0.06; *p* = 0.50	0.02; *p* = 0.80	0.04; *p* = 0.75	0.04; *p* = 0.72
**Supermarket Density 1 mile**	−0.02; *p* = 0.83	−0.03; *p* = 0.80	−0.03; *p* = 0.75	0.11; *p* = 0.26
**Supermarket Density 3 mile**	−0.07; *p* = 0.41	0.17; *p* = 0.11	0.14; *p* = 0.16	0.23; *p* = .02
**Supermarket Density 5 mile**	−0.01; *p* = 0.94	0.09; *p* = 0.39	0.07; *p* = 0.52	0.15; *p* = 0.15

**Table 3 nutrients-10-01974-t003:** Unadjusted regression analyses, with skin carotenoids as the dependent variable, and distance measures as independent variables.

Dependent Variable	Independent Variable	Parameter Estimate	Standard Error	*t*-Value	*p*-Value	*r*-Squared
Carotenoids (RS Device)	Distance to supermarket where surveyed (inverse)	−7.6	4.9	−1.53	0.13	0.2
Carotenoids (RS Device)	mRFEI (linear)	1.26	0.53	2.38	0.02 *	0.05
Carotenoids (RS Device)	mRFEI Healthy Outlets (cubic)	0.55	0.30	1.82	0.07	0.04
Carotenoids (RS Device)	mRFEI Unhealthy Outlets (cubic)	−0.005	0.003	−1.651	0.10	0.04
Carotenoids (RS Device)	mRFEI Local Service Area 1 mile (quadratic)	2.21 × 10^−2^	7.23 × 10^−3^	3.05	0.003 **	0.12
Carotenoids (RS Device)	mRFEI Local Service Area 3 mile (quadratic)	0.04	0.02	1.971	0.05 *	0.06
Carotenoids (RS Device)	mRFEI Local Service Area 5 mile (cubic)	−0.01	0.02	−0.618	0.54	0.04
Carotenoids (RS Device)	Supermarket Density 1 mile (quad)	1.596 × 10^−6^	2.272 × 10^−6^	0.703	0.49	0.009
Carotenoids (RS Device)	Supermarket Density 3 mile (cubic)	0.2396	0.2189	1.095	0.276	0.03
Carotenoids (RS Device)	Supermarket Density 5 mile (cubic)	−0.005455	0.025213	−0.216	0.829	0.01417
* *p*-value ≤ 0.05; ** *p*-value ≤ 0.01						

**Table 4 nutrients-10-01974-t004:** Adjusted regression analysis, with skin carotenoids as the dependent variable and mRFEI Localized Service Area 1 and 3 miles as independent variables.

Independent Variables	Estimate	Standard Error	*t*-Value	*p*-Value
Intercept	266.5	30.5	8.75	2.68 × 10^−14^
Age	−0.26	0.60	−0.44	0.66
Gender	−46.3	19.2	−2.4	0.01 **
Race	−34.5	41.7	−0.83	0.41
Outlet	−19.02	16.9	−1.13	0.26
WIC	55.4	25.3	2.19	0.03 *
SNAP	−13.24	17.5	−0.76	0.45
Reasons for Shopping: Near Home excluding near Work	2.9	16.7	0.18	0.86
mRFEI Local Service Area 1 mile	−6.02	173.28	−0.04	0.97
mRFEI Local Service Area 3 mile	2.4	0.79	3.08	0.003 **
Supermarket Distance (miles) (inverse)	−5.34	4.91	−1.09	0.28
(*F* = 2.186, *p* = 0.03), with an *R*^2^ of 0.26 * *p*-value ≤ 0.05; ** *p*-value ≤ 0.01				

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
