# Peer review of "Association between Spatial Access to Food Outlets, Frequency of Grocery Shopping, and Objectively-Assessed and Self-Reported Fruit and Vegetable Consumption"

_nutrients, 2018, doi:10.3390/nu10121974_

Reviewer 1 Report

The paper is about association between accesibility to the SM where surveyed, frequency of grocery shopping, and self- reported and objectively-assassed fruit and vegetable consumption. They also generated the proportions for sex, race, income, participation in SNAP and WIC.

There are some points to be addressed:

1) The title (Associations between objectively-assessed spatial access to food venues, frequency of grocery shopping, fruit and vegetable purchase, and consumption among eastern North Carolina supermarket customers) is better to be modified to (Associations between spatial access to food venues, frequency of grocery shopping, and objectively-assessed and self-reported fruit and vegetable consumption).

2) Line 167, "once per week or less should be deleted as  this frequency of shopping is mentioned in other categories.

3) Line 254, please clarify that these data are related to which of the tables.

4) Line 256, p<0.001 should be modified to p<0.05, as the p value is 0.002 that is more than 0.001. Also I suggest to mention the table 2 at the end of the line 256.

5) Line 261, figure 1, part c is the only figure for store type, so the legend of this figure should be modified and letter of d should be deleted for store type. I'm not sure if you have any figure that can show the correlation between skin carotenoid score and distance to SM where surveyed in(minutes) by store type, if you have please add it.

In figure 1 the font of the axis titles and legend is too small and should be changed to the larger one.

6) The legend of the table 4 should be modified as this table is about adjusted regression analysis with skin carotenoids as the dependent variable and independent variables.

7) Discussion is too brief and just cited to 8 papers. The author should work much more on this part of the manuscript. Also I didn't read anything about gender and frequency of shopping in discussion section.

8) There are many typo that should be fixed.

Line 260 surveyedin should be fixed to surveyed in.

Line 270 undadjusted should be fixed to unadjusted. depedent to dependent

Line 282 depedent should be fixed to dependent.

Author Response

The title   (Associations between objectively-assessed spatial access to food outlets, frequency   of grocery shopping, fruit and vegetable purchase, and consumption among   eastern North Carolina supermarket customers) is better to be modified to (Associations between spatial access to food outlets,   frequency of grocery shopping, and objectively-assessed and self-reported   fruit and vegetable consumption).

We have changed the   title based on the recommendation.

2) Line 167,   "once per week or less should be deleted as  this frequency of shopping is mentioned in   other categories.

This was deleted in   the manuscript per recommendations

3) Line 254, please   clarify that these data are related to which of the tables.

Clarified which it was   related to.

4) Line 256,   p<0.001 should be modified to p<0.05, as the p value is 0.002 that is   more than 0.001. Also I suggest to mention the table 2 at the end of the line   256.

Modified to P<.05.   and added mention.

5) Line 261, figure 1,   part c is the only figure for store type, so the legend of this figure should   be modified and letter of d should be deleted for store type. I'm not sure if   you have any figure that can show the correlation between skin carotenoid   score and distance to SM where surveyed in(minutes) by store type, if you   have please add it.

We modified the   legend. We added the figure you requested.

In figure 1 the font   of the axis titles and legend is too small and should be changed to the   larger one.

In order to increase   the font size, we had to increase the graphs and overall figure size, as   there wasn’t room for increased font otherwise. We have made this change.   Please let us know if we need to make additional changes.

6) The legend of the table 4 should be modified as this table is about   adjusted regression analysis with skin carotenoids as the dependent variable   and independent variables.

Changed to the   following:

“Adjusted regression analysis, with skin carotenoids as   the dependent variable and mRFEI Localized Service Area 1 and 3 miles as independent   variables.”

Discussion is too   brief and just cited to 8 papers. The author should work much more on this   part of the manuscript. Also I didn't read anything about gender and   frequency of shopping in discussion section.

We have made   substantial additions to the Discussion, further exploring topics brought up by   both reviewers. We have specifically addressed our findings around frequency   and gender.

“In our sample, men   had higher skin carotenoids levels than women. This finding is inconsistent   with previous research suggesting that women have higher skin carotenoid   status than men based on typically higher rates of fruit and vegetable   consumption and smaller body size, and elevated plasma carotenoid levels due   to physiological differences between the genders. One possible reason for our   findings was a statistically significant lower BMI (p=.006) for males   (avg=29) versus females (avg.=34) in our sample. Previous research has found   a negative association between BMI and total plasma carotenoid level   concentration due to storage/accumulation of carotenoids in adipose tissue.   Skin and plasma carotenoid levels have been found to correlate, although skin   may be a longer-term measure due to increased rates of depletion in plasma.   This finding should be further explored in future research.”

“We found no statistically   significant differences in skin carotenoid levels across levels of frequency   of shopping. Previous research in a similar population also found no significant   associations between frequency of shopping and fruit and vegetable   consumption for both supermarkets and farmers markets. Less frequent shopping   would seemingly be an indicator of less frequent fruit and vegetable   consumption due to the limited shelf life and increased rate of spoilage of   produce, but it may be that less frequent shoppers are compensating for   increased distance to the store through purchasing frozen and canned fruits   and vegetables. A high percentage of the sample shopped at the supermarket   more than once per week, so it could be that our sample had insufficient   representation from less frequent shoppers. Future research should further   explore this finding..”

8) There are many typo   that should be fixed.

Line 260 surveyedin   should be fixed to surveyed in.

Line 270 undadjusted   should be fixed to unadjusted. depedent to dependent

Line 282 depedent   should be fixed to dependent.

We have fixed these   and other typos.

Reviewer 2 Report

Comments to authors:

This paper aims to examine the association between fruit and vegetable consumption and access to supermarkets.  It introduces a novel objective measure of evaluation fruit and vegetable consumption in addition to self-reported intake.  Overall, this is a valuable addition to the field of food environments research, however I have some suggestions which may strengthen the paper.

The first suggestion relates to the CDC's mRFEI.  Is it overly simplistic to equate supermarkets and so on with 'healthy' food environments and small grocery stores etc. with 'less healthy' food environments?  Could this contribute to the mixed results from the studies summarised, and with the findings of your study?  Surely supermarkets are a main source of less healthy foods, as well as fruit and vegetables.  Certainly this is the case in Australia, where over half of packaged foods have been classified as unhealthy, and most sugar-sweetened beverages are purchased in supermarkets.  Perhaps a more sophisticated measure of the healthfulness of food environments is needed?  How would this impact your findings?

The second suggestion relates to the Veggie Meter tool, which I have not come across before.  More information is needed in the methods to highlight the novel value of this tool which provides an objective measure.  How has it been validated?  Was there a procedure to validate the data collected in this study?  Did the test impact your ability to recruit study participants?

The next suggestion relates to giving more attention to the reasons why participants’ selected the supermarket, given this study's findings that it often wasn't the closest to their homes.  What information did they provide in the survey?  What are the theories of the authors?  Could any of these reasons influence the amount of fruit and vegetables consumed?  Is there a need for further research?

The final suggestion related to food insecurity, which was mentioned as prevalent in the locations where this study took place.  How might this study’s findings inform policy to address food insecurity, including SNAP and WIC?  Selection of supermarket may influence the impact of these schemes, so can this study’s findings suggest ways to improve access and purchase of fruit and vegetables?

Minor comments:

L36 – abbreviation of supermarket to ‘SM’ seems unnecessary, and its also not appropriate to use the abbreviation in references.

L42 – it may be a cultural norm in the US that isn’t present in Australia, but I am more familiar with use of the term ‘food outlet’ in the literature, rather than ‘food venue’.

L42 – the word ‘resources’ isn’t needed

L46 – proximity to food venues, do you mean access to food venues?

L67 – what is the ‘stroke belt’?

L87-94 – please reword, as this sentence is currently not clear

L121- this section should be moved to the results, and information relating to how participants were surveyed placed here instead.

L142 – should this be added as a reference rather than a weblink in text?

L304 – as above, what are these factors?  How might they influence selection of supermarket?  Did participants also purchase food at other outlets?  Was this their main or top-up shop?

Author Response

Reviewer

Reviewer Comment

Response

Reviewer 1

The title   (Associations between objectively-assessed spatial access to food outlets, frequency   of grocery shopping, fruit and vegetable purchase, and consumption among   eastern North Carolina supermarket customers) is better to be modified to (Associations between spatial access to food outlets,   frequency of grocery shopping, and objectively-assessed and self-reported   fruit and vegetable consumption).

We have changed the   title based on the recommendation.

2) Line 167,   "once per week or less should be deleted as  this frequency of shopping is mentioned in   other categories.

This was deleted in   the manuscript per recommendations

3) Line 254, please   clarify that these data are related to which of the tables.

Clarified which it was   related to.

4) Line 256,   p<0.001 should be modified to p<0.05, as the p value is 0.002 that is   more than 0.001. Also I suggest to mention the table 2 at the end of the line   256.

Modified to P<.05.   and added mention.

5) Line 261, figure 1,   part c is the only figure for store type, so the legend of this figure should   be modified and letter of d should be deleted for store type. I'm not sure if   you have any figure that can show the correlation between skin carotenoid   score and distance to SM where surveyed in(minutes) by store type, if you   have please add it.

We modified the   legend. We added the figure you requested.

In figure 1 the font   of the axis titles and legend is too small and should be changed to the   larger one.

In order to increase   the font size, we had to increase the graphs and overall figure size, as   there wasn’t room for increased font otherwise. We have made this change.   Please let us know if we need to make additional changes.

6) The legend of the table 4 should be modified as this table is about   adjusted regression analysis with skin carotenoids as the dependent variable   and independent variables.

Changed to the   following:

“Adjusted regression analysis, with skin carotenoids as   the dependent variable and mRFEI Localized Service Area 1 and 3 miles as independent   variables.”

Discussion is too   brief and just cited to 8 papers. The author should work much more on this   part of the manuscript. Also I didn't read anything about gender and   frequency of shopping in discussion section.

We have made   substantial additions to the Discussion, further exploring topics brought up by   both reviewers. We have specifically addressed our findings around frequency   and gender.

“In our sample, men   had higher skin carotenoids levels than women. This finding is inconsistent   with previous research suggesting that women have higher skin carotenoid   status than men based on typically higher rates of fruit and vegetable   consumption and smaller body size, and elevated plasma carotenoid levels due   to physiological differences between the genders. One possible reason for our   findings was a statistically significant lower BMI (p=.006) for males   (avg=29) versus females (avg.=34) in our sample. Previous research has found   a negative association between BMI and total plasma carotenoid level   concentration due to storage/accumulation of carotenoids in adipose tissue.   Skin and plasma carotenoid levels have been found to correlate, although skin   may be a longer-term measure due to increased rates of depletion in plasma.   This finding should be further explored in future research.”

“We found no statistically   significant differences in skin carotenoid levels across levels of frequency   of shopping. Previous research in a similar population also found no significant   associations between frequency of shopping and fruit and vegetable   consumption for both supermarkets and farmers markets. Less frequent shopping   would seemingly be an indicator of less frequent fruit and vegetable   consumption due to the limited shelf life and increased rate of spoilage of   produce, but it may be that less frequent shoppers are compensating for   increased distance to the store through purchasing frozen and canned fruits   and vegetables. A high percentage of the sample shopped at the supermarket   more than once per week, so it could be that our sample had insufficient   representation from less frequent shoppers. Future research should further   explore this finding..”

8) There are many typo   that should be fixed.

Line 260 surveyedin   should be fixed to surveyed in.

Line 270 undadjusted   should be fixed to unadjusted. depedent to dependent

Line 282 depedent   should be fixed to dependent.

We have fixed these   and other typos.

Reviewer 2

The   first suggestion relates to the CDC's mRFEI.    Is it overly simplistic to equate supermarkets and so on with 'healthy'   food environments and small grocery stores etc. with 'less healthy' food   environments?  Could this contribute to   the mixed results from the studies summarised, and with the findings of your   study?  Surely supermarkets are a main   source of less healthy foods, as well as fruit and vegetables.  Certainly this is the case in Australia,   where over half of packaged foods have been classified as unhealthy, and most   sugar-sweetened beverages are purchased in supermarkets.  Perhaps a more sophisticated measure of the   healthfulness of food environments is needed?    How would this impact your findings?

We added the following   to the Discussion:

“Our findings suggest   a high level of complexity in the relationship between the food environment   and fruit and vegetable consumption, as displayed in the curvilinear   relationship between the two variables, which appears to be possibly be   explained by the combined influence of objective and subjective/attitudinal   factors. Consumers did not typically shop at the supermarket closest to their   home, but were influenced by a more complex food environment composition as   indicated by mRFEI, and by concepts like ‘acceptability’, ‘accommodation’,   and ‘cultural capital’. Access to a supermarket also didn’t necessarily lead   to healthier behaviors, which could be the result of exposure to store level   less healthy micro-environments or residing in neighborhoods with an overall   lower socioeconomic gradient. Given these findings, existing measurement   approaches, including the mRFEI, may be too simplistic given the complex   nature of food shopping and food purchasing decisions. There is a need for a   more complex and hyper-contextual composite measure of the food environment   that factors in objective measures of the food environment (distance, food   store mix, price, availability) along with more subjective measures that can   capture concepts like ‘acceptability’, ‘accommodation’, and ‘cultural   capital’. To support that, there may be a need to diversify the way we   conceive and measure the built food environment, taking into account the   physical composition of the retail environment where the food store resides   (e.g. retail density), localized sociodemographic variables (e.g. age,   income), and traffic patterns. Similarly, there is a need to diversify the   way that subjective/attitudinal factors are measured to better capture this   important component. With the vast availability of secondary data now   available to examine these hyper-contextual factors, efforts should be made   to determine how to best obtain and aggregate the data, and then configure a   composite score that factors in all of these measures.”

The second suggestion   relates to the Veggie Meter tool, which I have not come across before.  More information is needed in the methods   to highlight the novel value of this tool which provides an objective   measure.  How has it been   validated?  Was there a procedure to   validate the data collected in this study?    Did the test impact your ability to recruit study participants?

We added the following   the Methods section:

The Veggie Meter™ is   was found to be a valid measure of fruit and vegetable consumption among a   racially diverse sample in eastern North Carolina [60], with a correlation   coefficient of 0.71 (p<0.0001) between plasma carotenoids assessed via   High Performance Liquid Chromatography (the gold standard) and skin   carotenoids assessed via the Veggie Meter™

“Previous research has   found a negative association between BMI and total plasma carotenoid level   concentration due to storage/accumulation of carotenoids in adipose tissue.   Skin and plasma carotenoid levels have been found to correlate, although skin   may be a longer-term measure due to increased rates of depletion in plasma.   This finding should be further explored in future research.”

The next suggestion   relates to giving more attention to the reasons why participants’ selected   the supermarket, given this study's findings that it often wasn't the closest   to their homes.  What information did   they provide in the survey?  What are   the theories of the authors?  Could any   of these reasons influence the amount of fruit and vegetables consumed?  Is there a need for further research?

We have addressed this   topic in the Discussion section.

“  In our study, the two supermarkets where   customers were recruited are discount supermarkets located in commercial   areas in proximity to lower-income areas, which may potentially explain why   immediate proximity may be related to lower consumption, with a rise in   consumption between 3-5 miles and 5 to 7 minutes, and subsequent decrease   then leveling as distance increased. There has been considerable research   indicating those with lower incomes but live in neighborhoods with middle to   higher income block groups tend to have better health outcomes and report   better dietary intake. Given that these customers were selected from similar   stores in similar neighborhoods, the effect that distance has on dietary   intake may signify the neighborhood effects rather than simply access to a   supermarket. Thus, consideration of the geographic context of the stores   being sampled, including sociodemographic factors, may be important in future   research.

“This combination of   price and accessibility has been found to be a prominent factor in food   shopping, particularly for lower-income individuals, in previous research   (CITE). The prominence of price may also partially explain the skin   carotenoid levels among the participants in this study, if cost conscious   shoppers deem them too expensive relative to other foods. The prominence of   other factors may also explain why those closest the supermarket where they   shopped had the lowest skin carotenoid levels.”

“Most of the other   prominent factors, expressed nearly as frequently as price and distance, may   best be categorized under the domain of “acceptability”, defined as “people's   attitudes about attributes of their local food environment, and whether or   not the given supply of products meets their personal standards”, or the   domain of “accommodation”, defined as “how well food sources accept and adapt   to resident’s needs” (CITE). The concept of ‘cultural capital’, which refers   to how the subjective aspects of the store including ‘climate’ or ‘mood’   align with the customer’s self-image, may also be influencing shopping   decisions (CITE). Research has found that supermarkets are class-stratified,   where shoppers from different social classes shop at the outlet that caters   to their needs, suggesting that customers want a differentiated environment where   they are around customers who resemble themselves (CITE). Perceptions of   store level ‘acceptability’ may be a prominent factor determining store and   item choice, as research looking at supermarket shopping and social class   have found an “isn’t for me” perception. Measurement of ‘acceptability’ and   ‘accommodation’ of the food environment has been limited and underdeveloped   in the literature, thus further research is needed.”

“Our findings suggest   a high level of complexity in the relationship between the food environment   and fruit and vegetable consumption, as displayed in the curvilinear   relationship between the two variables, which appears to be possibly be   explained by the combined influence of objective and subjective/attitudinal   factors. Consumers did not typically shop at the supermarket closest to their   home, but were influenced by a more complex food environment composition as   indicated by mRFEI, and by concepts like ‘acceptability’, ‘accommodation’,   and ‘cultural capital’. Access to a supermarket also didn’t necessarily lead   to healthier behaviors, which could be the result of exposure to store level   less healthy micro-environments or residing in neighborhoods with an overall   lower socioeconomic gradient. Given these findings, existing measurement   approaches, including the mRFEI, may be too simplistic given the complex   nature of food shopping and food purchasing decisions. There is a need for a   more complex and hyper-contextual composite measure of the food environment   that factors in objective measures of the food environment (distance, food   store mix, price, availability) along with more subjective measures that can   capture concepts like ‘acceptability’, ‘accommodation’, and ‘cultural   capital’. To support that, there may be a need to diversify the way we conceive   and measure the built food environment, taking into account the physical   composition of the retail environment where the food store resides (e.g. retail   density), localized sociodemographic variables (e.g. age, income), and   traffic patterns. Similarly, there is a need to diversify the way that   subjective/attitudinal factors are measured to better capture this important   component. With the vast availability of secondary data now available to   examine these hyper-contextual factors, efforts should be made to determine   how to best obtain and aggregate the data, and then configure a composite   score that factors in all of these measures.”

The final suggestion   related to food insecurity, which was mentioned as prevalent in the locations   where this study took place.  How might   this study’s findings inform policy to address food insecurity, including   SNAP and WIC?  Selection of supermarket   may influence the impact of these schemes, so can this study’s findings   suggest ways to improve access and purchase of fruit and vegetables?

We have addressed this   topic in the Discussion section by adding the following:

“This concept may also   explain why access to supermarkets was not consistently associated with   healthier dietary behaviors, as microenvironments within supermarkets could   be considered less healthy environments, triggering negative food behaviors   for those exposed. Given this, our findings may support the use of policies   that encourage or create a healthier mix of food stores in the food   environment (considering the 3 mile localized road network buffer used in   this study), or modifications within the supermarket environment, to increase   healthier purchasing behaviors, including fruit and vegetable consumption.   Additionally, policy efforts aimed at improving food access should consider   our findings of supermarket choice, where shoppers  always did not shop at the supermarket   closest to their home. Thus, more careful and complex consideration should be   made when placing new stores in the food environment.”

“Our findings also   suggest that the impact of distance on fruit and vegetable consumption may   have been reduced with WIC participation. These findings support the   potential benefit of WIC participation in maintaining diet quality despite   reduced accessibility. The fact that WIC had a positive relationship with   skin carotenoid scores and SNAP had a negative (though not statistically   significant) relationship may be partially explained by the more restricted   foods allowed for purchase under the WIC program, which may encourage more   fruit and vegetable consumption. This aligns with recent research which found   low diet quality for SNAP participants compared to income-eligible   nonparticipants and higher-income individuals, potentially due to increased   levels of processed meat consumption.”

L36 – abbreviation of   supermarket to ‘SM’ seems unnecessary, and its also not appropriate to use   the abbreviation in references.

We changed all the   abbreviations to “supermarket (s)” and changed this in the references.

L42 – it may be a   cultural norm in the US that isn’t present in Australia, but I am more familiar   with use of the term ‘food outlet’ in the literature, rather than ‘food outlet’.

Changed “food outlets”   to “food outlets”

L42 – the word   ‘resources’ isn’t needed

Removed “resources”.

L46 –proximity to food   outlets, do you mean access to food outlets?

Replaced “proximity”   with “accessibility”

L67 – what is the   ‘stroke belt’?

Added: “US “Stroke Belt”   (a geographical region with elevated levels of cerebrovascular accident)”

L87-94 – please   reword, as this sentence is currently not clear

We reworded to the following:   “To understand the influence of the larger food environment, we used the   Reference USA business database (contains phone verified business listings)   to find food stores within the two counties where data collection was taking   place and all contiguous counties (to account for boundary effects). We extracted physical addresses for the following store   types: area chain supermarket (NAICS 445110; chains with 50 or more   employees), grocery stores (NAICS (445110; subcategorized as large (10-49   employees) and small (three or fewer)), supercenters (NAICS 452910),   wholesale clubs (NAICS 452910), convenience stores (445120), dollar stores   (452990), fruit and vegetable markets (NAICS 445230) and limited service   restaurants (722513).”

L121- this section   should be moved to the results, and information relating to how participants   were surveyed placed here instead.

The way the sentence was   worded made it seem like Results, but we were really hoping to the explain   the methods. We changed the wording to the following:

“Participants were   asked how often they shop at the outlet where they were surveyed, which   included options that ranged from “this is my first visit” to “six to seven   days a week”. They were also asked the reasons why they shopped at the   particular outlet, and the options they were given regarding accessibility included:   1) whether it was close to where they live, 2) worked, or 3) went to school,   or 4) if it was on their way to work home or school. Other options they were   given included: 1) them liking the food offered, 2) the store sells healthy foods,   3) they meet friends at the store, 4) the store has good prices/is   inexpensive, and 5) the store has good quality.”

L142 – should this be   added as a reference rather than a weblink in text?

We have changed it   from a weblink to a reference

L304 – as above, what   are these factors?  How might they   influence selection of supermarket?    Did participants also purchase food at other outlets?  Was this their main or top-up shop?

We added the following   to the Results to provide more information:

“Overall, accessibility   to their home was second in the top five reasons for shopping at the   supermarket. The top five most frequent reasons, in order of the number of   responses, were the following: 1) ‘It has good prices/is inexpensive’   (n=115), 2) ‘It is close to where I live’ (n=110), 3) ‘It sells the food I   like’ (n=109), 4) ‘It has good quality’ (n=106), 5a) ’It has good service/You   know the owner/staff are friendly and helpful’ (n=104), 5b) ‘It is clean’   (n=104).”

In the Discussion, the   following was added:

“ In our study, the   two supermarkets where customers were recruited are discount supermarkets   located in commercial areas in proximity to lower-income areas, which may   potentially explain why immediate proximity may be related to lower   consumption, with a rise in consumption between 3-5 miles and 5 to 7 minutes,   and subsequent decrease then leveling as distance increased. There has been   considerable research indicating those with lower incomes but live in   neighborhoods with middle to higher income block groups tend to have better   health outcomes and report better dietary intake. Given that these customers   were selected from similar stores in similar neighborhoods, the effect that   distance has on dietary intake may signify the neighborhood effects rather   than simply access to a supermarket. Thus, consideration of the geographic   context of the stores being sampled, including sociodemographic factors, may   be important in future research..”

“Overall, proximity to   the home was the second most prominent factor behind store price among the   reasons for shopping at the store, and shoppers also frequently mentioned the   factors of ‘selling food I like’, “it has good quality’, “good service’, and   ‘it is clean’. This suggests that proximity of the outlet to the home   matters, but other factors might influence outlet preference among those that   are proximal. This combination of price and accessibility has been found to   be a prominent factor in food shopping, particularly for lower-income   individuals, in previous research. The prominence of price may also partially   explain the skin carotenoid levels among the participants in this study, if   cost conscious shoppers deem them too expensive relative to other foods. The importance   of non-spatial factors may also explain why those closest the supermarket   where they shopped had the lowest skin carotenoid levels.

Most of the other   prominent factors, expressed nearly as frequently as price and distance, may   best be categorized under the domain of “acceptability”, defined as “people's   attitudes about attributes of their local food environment, and whether or   not the given supply of products meets their personal standards”, or the   domain of “accommodation”, defined as “how well food sources accept and adapt   to resident’s needs” (CITE). The concept of ‘cultural capital’, which refers   to how the subjective aspects of the store including ‘climate’ or ‘mood’   align with the customer’s self-image, may also be influencing shopping   decisions (CITE). Research has found that supermarkets are class-stratified,   where shoppers from different social classes shop at the outlet that caters   to their needs, suggesting that customers want a differentiated environment   where they are around customers who resemble themselves (CITE). Perceptions   of store level ‘acceptability’ may be a prominent factor determining store   and item choice, as research looking at supermarket shopping and social class   has found an “isn’t for me” perception. Measurement of ‘acceptability’ and   ‘accommodation’ of the food environment has been limited and underdeveloped   in the literature, thus further research is needed.”
